# Differences and Similarities in the Pattern of Early Metabolic and Morphologic Response after Induction Chemo-Immunotherapy versus Induction Chemotherapy Alone in Locally Advanced Squamous Cell Head and Neck Cancer

**DOI:** 10.3390/cancers14194811

**Published:** 2022-09-30

**Authors:** Michael Beck, Sabine Semrau, Marlen Haderlein, Antoniu-Oreste Gostian, Julius Hartwich, Sarina Müller, Annett Kallies, Carol-Immanuel Geppert, Miriam Schonath, Florian Putz, Udo Gaipl, Benjamin Frey, Marc Saake, Heinrich Iro, Michael Uder, Arndt Hartmann, Torsten Kuwert, Rainer Fietkau, Markus Eckstein, Markus Hecht

**Affiliations:** 1Clinic of Nuclear Medicine, Friedrich-Alexander-Universität Erlangen-Nürnberg, 91054 Erlangen, Bavaria, Germany; 2Department of Radiation Oncology, Friedrich-Alexander-Universität Erlangen-Nürnberg, 91054 Erlangen, Bavaria, Germany; 3Department of Otolaryngology—Head & Neck Surgery, Friedrich-Alexander-Universität Erlangen-Nürnberg, 91054 Erlangen, Bavaria, Germany; 4Institute of Pathology, Friedrich-Alexander-Universität Erlangen-Nürnberg, 91054 Erlangen, Bavaria, Germany; 5Department of Diagnostic Radiology, University Hospital of Erlangen, 91054 Erlangen, Bavaria, Germany

**Keywords:** response, immunochemotherapy, chemotherapy, ^18^F-FDG-PET/CT, MRI, computer tomography, progression, dissociative response, PERCIST, pseudoprogression

## Abstract

**Simple Summary:**

There are differences and similarities when assessing the short-term therapeutic response of chemotherapy and chemotherapy plus immunotherapy using imaging techniques, which may be necessary to make treatment decisions in malignant head and neck tumors. After both chemo- and immunochemotherapy, remission becomes measurable in cross-sectional and metabolic diagnostics after only one cycle of therapy, with ^18^FDG-PET/CT predicting complete remission of tumor cells in a representative biopsy better than MRI/CT examination in both therapeutic modalities. While complete tumor remission is highly likely (88%) after immunochemotherapy in tumors with low residual activity (≤40% of initial SUV), this is less common after chemotherapy alone (65%). In metabolic nonresponse with more than 80% residual activity, the probability of complete remission nevertheless is low after chemotherapy alone (6%). After immunochemotherapy, these false nonresponders are common (35%), requiring additional diagnostics by deep biopsy. Cases of pseudoprogression with an increase of SUV_max_ of more than 125% of the baseline were not observed.

**Abstract:**

**Background:** In head and neck cancer patients, parameters of metabolic and morphologic response of the tumor to single-cycle induction chemotherapy (IC) with docetaxel, cis- or carboplatin are used to decide the further course of treatment. This study investigated the effect of adding a double immune checkpoint blockade (DICB) of tremelimumab and durvalumab to IC on imaging parameters and their significance with regard to tumor cell remission. **Methods:** Response variables of 53 patients treated with IC+DICB (ICIT) were compared with those of 104 who received IC alone. Three weeks after one cycle, pathologic and, in some cases, clinical and endoscopic primary tumor responses were evaluated and correlated with a change in 18F-FDG PET and CT/MRI-based maximum-standardized uptake values (SUV_max_) before (SUV_max_pre), after treatment (SUV_max_post) and residually (resSUV_max_ in % of SUV_max_pre), and in maximum tumor diameter (D_max_) before (D_max_pre) and after treatment (D_max_post) and residually (resD). **Results:** Reduction of SUV_max_ and D_max_ occurred in both groups; values were SUV_max_pre: 14.4, SUV_max_post: 6.6, D_max_pre: 30 mm and D_max_post: 23 mm for ICIT versus SUV_max_pre: 16.5, SUV_max_post: 6.4, D_max_pre: 21 mm, and D_max_post: 16 mm for IC alone (all *p* < 0.05). ResSUV_max_ was the best predictor of complete response (IC: AUC: 0.77; ICIT: AUC: 0.76). Metabolic responders with resSUV_max_ ≤ 40% tended to have a higher rate of CR to ICIT (88%; n = 15/17) than to IC (65%; n = 30/46; *p* = 0.11). Of the metabolic nonresponders (resSUV_max_ > 80%), 33% (n = 5/15) achieved a clinical CR to ICIT versus 6% (n = 1/15) to IC (*p* = 0.01). **Conclusions:** ICIT and IC quickly induce a response and 18F-FDG PET is the more accurate modality for identifying complete remission. The rate of discrepant response, i.e., pCR with metabolic nonresponse after ICIT was >30%.

## 1. Introduction

Positron emission tomography or computed tomography (PET/CT) has become an established diagnostic imaging modality for evaluating early treatment response in patients with certain tumor entities, e.g., Hodgkin’s lymphoma [1,2] and esophageal cancer [3]. PET/CT can also be used to assess the early response of head and neck cancer (HNC) to chemotherapy as various studies have found it effective in identifying HNC patients who will achieve long-term response to chemoradiotherapy (CRT) [4,5,6]. This applies to tumors in all sites that typically make it necessary to weigh the pros and cons of an organ-sparing approach against those of surgical resection [7]. If an early treatment response is detected, some research groups believe that the administration of one cycle of induction chemotherapy (IC) is sufficient to decide whether to proceed with CRT or surgery [8,9]. This is particularly interesting since it has been shown that additional chemotherapy does not improve survival and, furthermore, that additional treatment cycles and agents lead to an increase in toxicity and death rates [10].

Current efforts are aimed at modifying induction treatment to identify patients who will benefit from immune checkpoint blockade (ICB) in conjunction with radiotherapy. One new approach consists of adding immune checkpoint inhibitor therapy (ICIT) to the short induction phase. Instead of CRT, responders to this protocol receive radiation plus antibody therapy with the goal of reducing the unfavorable long-term functional impacts of platinum-containing CRT [11,12], whereas nonresponders still receive CRT so as not to jeopardize their chances of recovery. The present study investigates the effect of expanding the induction treatment protocol to include double immune checkpoint blockade (DICB) with a combination of programmed cell death protein ligand 1 (PD-L1) and cytotoxic T-lymphocyte-associated protein 4 (CTLA-4). Responders to this protocol are currently predicted by determining tumor-infiltrating CD8+ T cell density, for which two representative deep biopsies are needed. The invasive nature of biopsy and the quality of the collected specimens are relevant issues that raise the question of whether patients should still be identified as responders and nonresponders to induction treatment based on morphologic criteria [13] or by imaging criteria.

In addition to head and neck tumors, combination chemoimmunotherapy protocols are increasingly becoming the standard of care for solid tumor patients. Early identification of responders and nonresponders is useful for two reasons: first, it may eliminate additional treatment toxicity and intensity by identifying patients who are unlikely to mount an adequate response and, second, it can encourage patients to continue treatment when they are experiencing side effects but are likely to respond in the long term.

In a previous study in a small cohort, we found that the early metabolic response to immunochemotherapy is sensitive to achieving pathological remission to one cycle therapy [14]. These results should be validated on a larger number of patients with locally advanced head and neck squamous cell cancer (HNSCC) and compared with other imaging methods as well as compared with our observations in patients who received a single shot chemotherapy only.

## 2. Materials and Methods

### 2.1. Patients and Treatments (Figure 1)

A prospective cohort of 53 patients treated with induction chemoimmunotherapy (ICIT) consisting of a single cycle of induction chemotherapy and a double immune checkpoint blockade therapy in the scope of the CheckRad-CD8 study was compared with a retrospective cohort of 104 patients treated with single-cycle induction chemotherapy from 2008–2020. Before and three weeks after treatment, all patients’ primary tumors were assessed by contrast-enhanced computed tomography (CT) or magnetic resonance imaging (MRI), and 51 of the 53 ICIT patients were additionally evaluated by ^18^F-fluoro-D-glucose (^18^F-FDG) uptake positron emission tomography (PET)/computed tomography (^18^F-FDG PET/CT). All imaging studies were performed on scanners by Siemens Healthineers (Erlangen, Germany).

Induction chemotherapy was with one cycle of docetaxel (75 mg/m^2^) on day 1 plus cisplatin (30 mg/m^2^) or carboplatin (AUC 1.5) on days 1–3. CheckRad-CD8 study treatment comprised ICIT with the aforementioned IC protocol plus DICB with durvalumab (1500 mg absolute) and tremelimumab (75 mg) on day 5. ICIT patient were studied by prospective data collection.

### 2.2. CT/MRI Response Evaluation

Depending on which method was available, MRI or contrast-enhanced CT was used to measure the morphologic response to treatment with one cycle of IC or ICIT, defined as the change in maximum tumor diameter from before treatment (D_max_pre) to 21–28 days after treatment (D_max_post) according to RECIST, but limited to the primary tumor. Diameter measurements were obtained in independent CT or MRI studies as well as in the CT component of PET/CT. The decrease or increase in tumor diameter was defined as the residual diameter (resD), which was calculated as a percentage of baseline as follows: resD_max_ (%) = (D_max_post/D_max_pre) × 100. An assessment of the response of the lymph nodes was waived because it did not play a role in the therapy decision and no tissue samples were obtained from the lymph nodes after the induction phase.

### 2.3. Metabolic Response Evaluation

Metabolic response of the primary tumor was defined as the change in ^18^F-FDG PET and CT/MRI-based maximum standardized uptake values (SUV_max_) in the period before (SUV_max_pre) and after treatment (SUV_max_post). The decrease or increase in the residual tumor activity (resSUV_max_) was likewise calculated as a percentage of baseline: resSUV_max_ (%) = (SUV_max_post/SUV_max_pre) × 100. The level of response was graded according to the method of [9], where resSUV_max_ ≤ 40% is defined as a high-metabolic response, resSUV_max_ > 40% but <80% as a moderate metabolic response, and resSUV_max_ > 80% as a metabolic nonresponse. Further remission criteria of the EORTC [15] and according to [14] were evaluated for predictive accuracy.

### 2.4. Clinical and Pathological Response Evaluation

Endoscopy was performed before and 21–28 days after the start of induction chemotherapy for response evaluation purposes. Pathologic complete response (CR) was defined as the absence of tumor cells in a representative deep extensive excision biopsy specimen from the initial and subsequent PET-positive primary tumor region performed in all cases of the ICIT patients and some of the IC patients. Endoscopic complete response, defined as the absence of a scar after induction treatment in panendoscopy studies by two examiners, was used as an alternative measure of complete response in some cases of IC patients for whom biopsy samples were unavailable or not clearly representative of the former tumor region. The study flowchart can be found in Figure 1.

**Figure 1 cancers-14-04811-f001:**
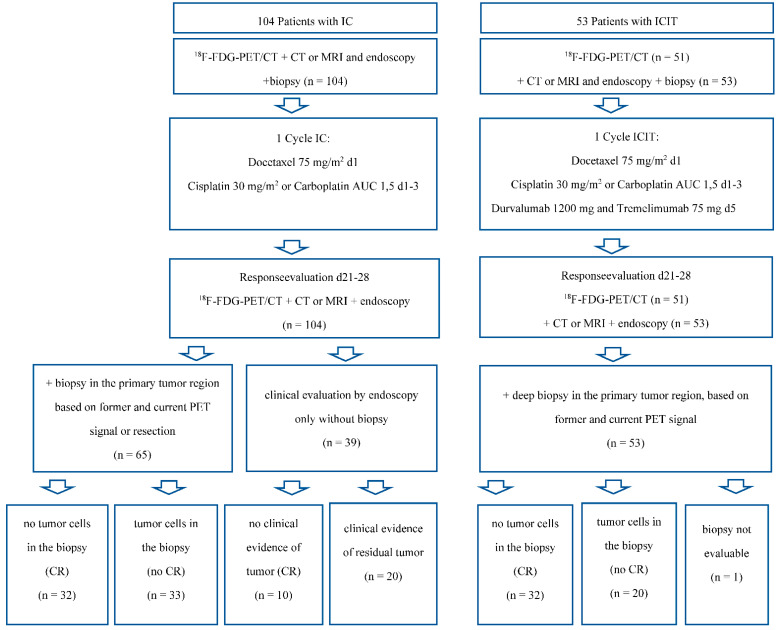
Patients, treatment, and response evaluation.

### 2.5. Statistical Analysis

Chi-square and *t*-tests were used to compare frequencies and to test for differences in dependent and independent variables, respectively, between the two groups (ICIT vs IC). Receiver operating characteristic curve (ROC) analysis was performed to compare the sensitivity and specificity of individual variables to predict complete remission. All statistical analyses were performed with SPSS Statistics Version 28 (IBM, Armonk, NY, USA).

## 3. Results

### 3.1. Patients

The ICIT and IC cohorts were comparable in terms of gender, age, and N stage distribution. However, an imbalance of the T-stage, UICC stage, tumor-grade distribution, and the proportion of HPV-associated OPSCC was observed. Specifically, the ICIT cohort contained more patients with higher-stage tumors and more human papilloma virus (HPV)-associated oropharyngeal squamous cell carcinoma (OPSCC) patients (Table 1).

### 3.2. Change in Tumor Diameter after IC versus ICIT

One cycle of treatment resulted in tumor regression in both groups. The maximum tumor diameter decreased from 30 ± 14 mm to 23 ± 16 mm after ICIT, and from 21 ± 9 mm to 16 ± 10 mm after IC (*p* < 0.001 for both), whereby ICIT patients had significantly larger tumors at baseline (*p* = 0.024). ICIT showed a tendency toward a smaller residual tumor size (resD_max_) relative to the baseline compared to IC (71 ± 31 % vs 78 ± 19 %, *p* = 0.073). The resD_max_ after IC and ICIT did not depend on the imbalanced distributed parameters such as the T-stage (*p* = 0.887, *p* = 0.077), UICC-stage (*p* = 0.397, *p* = 0.459), tumor grade (*p* = 0.761, *p* = 0.848), and showed no difference between HPV pos OPSCC and other tumors (*p* = 0.939, *p* = 0.206) except for tumor localization (*p* = 0.036 in the IC group, *p* = 0.765 in the ICIT).

### 3.3. Metabolic Response Rates after IC versus ICIT

Paired before–after ^18^F-FDG-PET studies were available for 153 out of 157 patients in the overall sample, specifically, for 104/104 IC patients and 51/53 ICIT patients. Primary tumor visualization was achieved in all cases. SUV_max_ also decreased significantly in both treatment arms, from 16.4 ± 9.4 to 6.4 ± 6.5 after IC (*p* < 0.001), and from 14.4 ± 7.9 to 6.6 ± 5.5 after ICIT (*p* < 0.001). In contrast to the residual tumor size, the residual metabolic activity (resSUV_max_) tended to be smaller after IC (42 ± 32%) than after ICIT (52 ± 37%) (*p* = 0.09).

The resSUV_max_ after IC and ICIT was not different between several T-stages (*p* = 0.105, *p* = 0.405), UICC-stages (*p* = 0.344, *p* = 0.934), between different grading (*p* = 0.149, *p* = 1.00), localizations (*p* = 0.774, *p* = 0.138) and showed no difference between HPV pos OPSCC and other tumors (*p* = 0.227, *p* = 0.271).

### 3.4. Clinical and Pathologic Response to ICIT versus IC

Of the 53 patients treated with ICIT, 32 (60.3%) had no detectable residual tumor in directed biopsy specimens from the former tumor site, and the remaining 21 had residual tumor (n = 20) or a nonevaluable sample (n = 1) (Figure 1). After the IC alone (n = 104), 32 patients had no histologically detectable residual tumor cells in representative biopsy specimens examined by panendoscopy and 10 had sufficient evidence of clinical complete response to dispense with biopsy, whereas 33 patients had residual tumor cells in biopsy specimens assessed by panendoscopy, and another 29 had macroscopic tumor persistence but no collected histology specimens. Contrary to the imaging results, pathologic CR rates were higher after ICIT than after IC (42/104 patients: 40.3%, *p* = 0.018).

### 3.5. Differences in Imaging Variables between CR and non-CR after ICIT versus IC

Imaging parameters are summarized by treatment and response characteristics in Table 2. CT/MRI studies for morphologic tumor response evaluation showed that the initial differences in tumor size between IC and ICIT at baseline were still present after treatment, resulting in comparable differences in percentage residual tumor size (resD_max_). This applies to both CR (*p* = 0.970) and non-CR (*p* = 0.218) in ICIT and IC. This means that at this early point in time it is not possible to differentiate between responders and nonresponders using morphological imaging.

Conversely, ^18^F-FDG PET/CT showed significant differences in resSUV_max_ between CR and non-CR patients of both groups, whereby CR was associated with a lower resSUV_max_ than non-CR (*p* < 0.001).

Complete responders to ICIT had a significantly higher residual SUV_max_ (41%) than complete responders to IC alone (28%, *p* = 0.025). This was also the case for non-CR patients, who had residual SUV_max_ values of 76% vs 53%, respectively (*p* = 0.032). Relatively speaking, ICIT with induction chemotherapy plus double-immune checkpoint blockade was associated with higher residual activity independent of CR or non-CR status.

Regarding the individual treatment groups, ROC analysis of ICIT (Figure 2A) and IC (Figure 2B) showed that metabolic parameters of response with resSUV_max_ (AUC 0.76 for ICIT, *p* < 0.001; AUC 0.77 for IC, *p* < 0.001) and SUV_max_post (AUC 0.80 for ICIT, *p* < 0.001; AUC 0.71 for IC, *p* < 0.001) were better predictors of pathological response to both treatment modalities than SUV_max_pre (AUC 0.42, AUC 0.60). CT and MRT-based variables did not prove to be suitable predictors of response to ICIT (D_max_pre: AUC 0.67, D_max_post: AUC 0.66, resD: AUC 0.57). After IC resD has an AUC of 0.72 (*p* < 0.001), but D_max_post and D_max_pre did not predict responses to IC well (AUC 0.57, AUC 0.53).

### 3.6. The Accuracy of Predicting CR to IC and ICIT Using Several Established Metabolic Values Based on resSUV_max_

Regarding the prognostic value of the various markers for predicting a complete response, a low resSUV_max_ (≤40%) [6,9] was shown to have a sensitivity, specificity, positive predictive value (PPV), and negative predictive value (NPV) of 71%, 74%, 67%, and 79% for IC compared to 90%, 48%, 88%, and 42% for ICIT. This indicates that in patients whose post-treatment SUV_max_ is less than 40% of the baseline, the odds of complete remission were higher after ICIT (88%) than after IC (65%). Patients after ICIT who achieve this value had a high probability of complete remission.

The positive predictive value for recognizing a complete remission based on the EORTC-criterion [15] is lower and the criterion (resSUVmax ≤ 50%) plus value of six as SUV_max_post does not produce a clinical important improvement in the prediction accuracy (Table 3).

### 3.7. Metabolic Nonresponse and Frequency of Unexpected CR (Discrepant Response)

Interestingly, 5/15 patients without metabolic response (resSUV_max_ ≥ 80% of baseline) had a complete pathological response without tumor cells in the biopsy with ICIT, which meant 5/51 patients. The number of patients having a discrepant response after ICIT was not different between patients with HPV-associated OPSCC 5% (1/20) and patients without HPV-associated OPSCC at 12% (4/31). In patients after chemotherapy, it was 1/16, which meant 1/104. Conversely, every third patient with metabolic nonresponse concealed a pathological respondent (discrepant responder) (see Figure 3a–c). However, no one had metabolic progression (resSUV_max_ >125%) [15] despite complete pathological remission (pseudoprogression).

## 4. Discussion

Immune checkpoint inhibitors (ICIs), which work by blocking insufficient T-cell-specific antitumor responses that maintain immune tolerance, have become an integral part of cancer management. They are used to treat a variety of solid tumor entities, mainly in a metastatic setting [16,17,18,19,20] but increasingly in the early stages as well [21,22]. The survival benefit of immune checkpoint inhibitors PD-1 and PD-L1 for head and neck cancer patients has also been demonstrated in phase III clinical trials [23,24]. PD-L1 expression in cancer cells and tumor-infiltrating lymphocytes was identified as a predictor of response [24]. Microsatellite instability, mismatch repair deficiency, and tumor mutational burden have been identified as biomarkers of treatment efficacy in other cancer types.

The KEYNOTE-048 study showed that in head and neck squamous cell cancer (HNSCC) patients with a PD-L1 combined positive score (CPS) of ≥1, the addition of immunotherapy to chemotherapy increased the odds of 24-month survival two-fold whereas in those with a CPS of ≥20, chemotherapy had no additional benefit [24]. The prognostic value of PD-L1 expression is, however, limited. Firstly, ICI therapy resulted in long-term progression-free survival in only 15 to 20 percent of patients with high-PD-L1 expression [24]. Moreover, patients with low PD-L1 expression are also responders, so PD-L1 expression cannot be used as a biomarker of response to ICI treatment for all tumor entities [19,23].

Therefore, it makes sense to look for other early predictive markers for example in the field of imaging. In this context, there is a scarcity of data on very early PET response to immune therapy and chemoimmunotherapy. However, there is some experience regarding early markers in imaging after one cycle of chemotherapy for predicting long-term treatment effects but also short-term responses [4,6,9,25]. When comparing the old and recent observations, there are similarities and differences after one cycle of chemoimmunotherapy compared to one cycle of chemotherapy alone. Even after one cycle of intensified induction of ICIT, metabolism was shown to decrease numerically more than morphological metrics. According to expectations, ICIT resulted in greater morphologic remission than after IC, although it should be noted that the two groups were not entirely comparable in terms of tumor size and other characteristics at the beginning of treatment. However, there were no differences in relative residual size or residual metabolism between the individual groups of T-stage, UICC-stage, or grading, suggesting that these imbalances had probably no great influence on the two important parameters resSUV_max_ and resD_max_. Although this is a weak point of this study, so far there are no groups of this size that are more suitable for the comparison.

Furthermore, it could be observed that, the decrease in metabolic activity was underproportional after ICIT versus IC as a sign of increased activity, possibly triggered by reactivation of the immune response. In addition, resSUV_max_ as a fraction of baseline was found to predict complete remission quite well compared with other parameters, but not as well as after IC. This makes the interpretation of PET results more difficult than after IC.

We observed that a persistently high-glucose metabolism according to ICIT could hide a significant remission. In our study, we considered the complete absence of tumor cells in an extensive biopsy as a reference. We would interpret this as the first sign of pseudoprogression. Regarding the PET response criteria [26,27], pseudoprogression is known to occur. This was considered when framing the immune response evaluation criteria in solid tumors (iRECIST) [28], but the wording of the criteria is still rather vague. Pseudoprogression rates ranging from 3% [29], 5% [30], and 12% [31] have been reported for various cancer monotherapies. In fact, we did not have anyone who met the criteria of pseudoprogression at this early stage, especially since we also lacked the PET course. However, we had a rate of more than 10% of patients with complete pathologic remission but metabolic nonresponse in our cohort of ICIT, which can lead to considerable irritation in treatment decision making. In contrast, only 1% of our chemotherapy patients had this constellation, which corresponds to the usual rate [27]. Pseudoprogression and immune dissociated response are attributed to immune cell infiltration [30,31]. The high rate was a remarkable finding, which could be the result of the quadruple treatment regimen including accelerated immune response triggered by cell death due to chemotherapy or the early assessment time point after the application of both therapies. However, the results are valid, especially in our ICIT patients, because they were obtained on the basis of biopsy findings. The biopsy specimens were taken from initially metabolically active and previously biopsied areas. In addition, two patients underwent repeat biopsies. The issue of how to interpret a persistently high-glucose metabolism in treatment decision making is still unsolved. A wait-and-see approach is unjustifiable in view of the negative impact that not recognizing true progression would have on the resectability of borderline resectable tumors and the adverse effect that any distant metastases developing in the interim could have on curability. Various PET studies, which have shown that a dissociated response pattern of involved lymph nodes compared to primary tumor suggests a clinical benefit of immunotherapy and a favorable prognosis [29,32], may curtail this problem [29]. So far, the problem of persistently high-PET signal despite clinical response can still only be addressed by biopsy and re-biopsy with the knowledge that such a condition is much more common than after chemotherapy alone. A clue may be a decision based on the infiltration of CD8+ cells [33].

Our study, on the other hand, showed that if there is a significant decrease in SUV_max_ in the primary tumor after induction treatment, which was the case in 40% of ICIT patients, there is a high probability of pathological complete response. The probability of the prediction being true was 88% and thus higher than that after chemotherapy. In this case, the patient does not need a deep biopsy of the tumor. Further followup studies are needed to demonstrate the extent to which early metabolic response parameters accurately predict the long-term success of treatment, specifically, if pseudoprogression and dissociated response are associated with a better prognosis [34,35]. However, there is no unanimous opinion on the frequency of activation of other lymphoid tissue-rich organs and their prognostic value in predicting the efficacy and outcome of immunotherapy [36,37], but in our population we did not observe such activation [29].

## 5. Conclusions

After induction chemoimmunotherapy, very early response evaluation is possible and is best performed by ^18^F-FDG PET/CT compared to morphological response. Residual SUV_max_ and post-treatment SUV_max_ showed good correlation with response. A resSUV_max_ of ≤40% of the baseline, observed in 40% of our patients, was associated with a very high probability of complete response and there was probably no further biopsy needed. In contrast, the situation with high-residual uptake (resSUV_max_ of >80%) was associated with complete remission in one-third of patients (false nonresponse). This was observed significantly less often after induction chemotherapy alone. Therefore, further identification criteria must be established in the latter case.

## Figures and Tables

**Figure 2 cancers-14-04811-f002:**
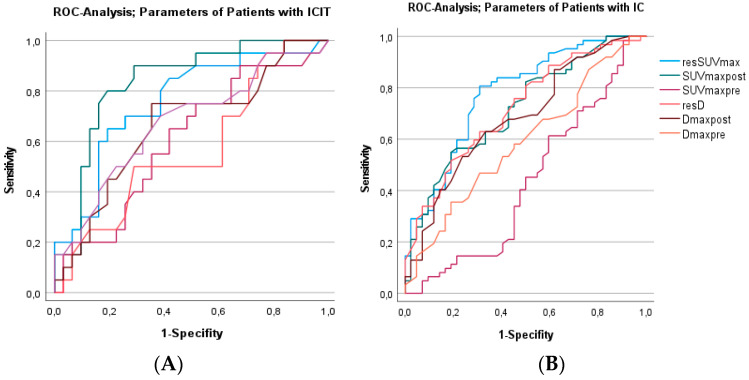
(**A**,**B**): ROC Analysis of different imaging parameters of 18F-FDG-PET/CT and morphologic imaging (CT/MRI) in patients after one cycle of ICIT ((**A**), left) and IC ((**B**), right). Sensitivity and specifity for the predefined cut off resSUVmax ≤ 40% ICIT: 90%, 48%, IC: 74%, 71%.

**Figure 3 cancers-14-04811-f003:**
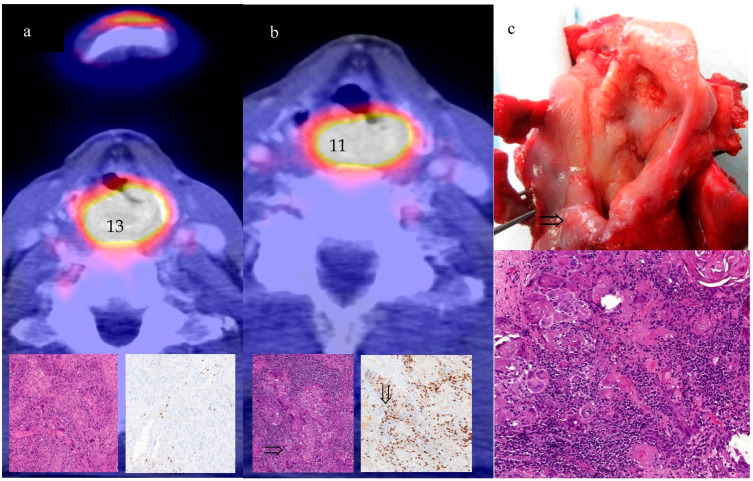
(**a**–**c**): supraglottic laryngeal cancer before induction (**a**) with SUV_max_pre: 13, HE staining: squamous cell cancer and sparse CD8+; (**b**) after induction chemotherapy plus durvalumab or Tremelimumab, SUV_max_post: 11, regressive changes but (⇒) still tumor cells and infiltration of CD8+ lymphocytes (⇓); (**c**) the larynx after laryngectomy with suspicion of persisting tumor (⇒) HE staining without any tumor cells but lymphocytes.

**Table 1 cancers-14-04811-t001:** Patient characteristics.

	IC Patients		ICIT Patients		*p*-Value
	(n)	(%)	(n)	(%)	(Chi^2^, * Fisher’s exact and ** independent samples *t*-test)
Patients	104		53		
Male	84	80.8	45	84.9	*p* = 0.66 *
Female	20	19.2	8	15.1	
**Median age** (range)	58 (35–78)		61 (38–78)		*p* = 0.27 **
**T stage**					*p* < 0.001
1	1	0.9	2	3.8	
2	35	33.7	7	13.2	
3	42	40.4	10	18.9	
4	26	25.0	34	64.2	
**N stage**					*p* = 0.803
0	32	30.7	14	26.4	
1	17	16.3	10	18.9	
2a	1	1.0	2	3.8	
2b	26	25.0	13	24.5	
2c	27	26.0	14	26.4	
3	1	1.0	0	0	
**UICC Stage (7th Edition)**					*p* = 0.025
2	12	11.5	0	0	
3	23	22.1	10	18.9	
4	69	66.4	43	81.1	
**Grade**					*p* = 0.016
1	4	3.8	0	0	
2	49	47.2	8	15.1	
3	46	44.2	24	45.3	
Missing, HPV-positive OPSCC	5	4.8	21	39.6	
**HPV-associated OPSCC**					*p* < 0.001 *
No	99	95.2	32	60.4	
Yes	5	4.8	21	39.6	
**Localization**					*p* < 0.001
Oral cavity/oropharynx	18	17.3	33	62.3	
Hypopharynx	42	40.4	11	20.8	
Larynx	44	42.3	9	17.0	

**Table 2 cancers-14-04811-t002:** Imaging results for morphologic and metabolic parameters of pathologic and clinical complete response (CR) and non-CR measured before (pre) and after (post) ICIT versus IC.

	Complete Response (CR)		Noncomplete Response (non-CR)		*p*-Value: CR versus non-CR to ICIT	*p*-Value: CR versus non-CR to IC
	ICIT (n = 31/32)	IC (n = 42)	ICIT (n = 20/21)	IC (n = 62)		
SUV_max_ pre	12.7 ± 7.7	17.5 ± 11.0	15.9 ± 8.1	15.7 ± 8.0	*p* = 0.213	*p* = 0.24
	*p* = 0.021		*p* = 0.901			
SUV_max_ post	4.8 ± 4.1	4.6 ± 4.3	10.6 ± 5.6	7.5 ± 7.2	***p* < 0.001**	***p* < 0.001**
	*p* = 0.828		*p* = 0.063			
Residual SUV_max_ (%)	41 ± 28	28 ± 20	76 ± 40	53 ± 33	***p* = 0.02**	***p* = 0.00**
	***p* = 0.025**		***p* = 0.032**			
D_max_ pre (mm)	28 ± 11	20 ± 8	33 ± 15	22 ± 10	*p* = 0.038	*p* = 0.13
	*p* = 0.003		*p* < 0.001			
D_max_ post (mm)	18 ± 14	13 ± 7	27 ± 16	18 ± 9	*p* = 0.056	*p* = 0.002
	*p* = 0.063		*p* = 0.023			
Residual D_max_ (%)	72 ± 34	70 ± 19	68 ± 24	81 ± 17	*p* = 0.47	*p* < 0.001
	*p* = 0.97		*p* = 0.218			

**Table 3 cancers-14-04811-t003:** PET response with different thresholds testing for CR.

PET Response and Clinical or Pathological Response				
	Sensitivity	Specifity	NPV	PPV
resSUVmax ≤ 75% (EORTC for metabolic response)				
ICIT	50%	72%	66%	72%
IC	26%	92%	94%	47%
**resSUVmax ≤ 40% (Semrau 2015, 2021) for metabolic response)**				
ICIT	90%	48%	42%	**88%**
IC	74%	71%	79%	**65%**
**resSUVmax ≤ 50% + SUVmaxpost < 6 (Beck 2022 for metabolic response)**				
ICIT	95%	45%	52%	**93%**
IC	71%	59%	72%	58%

## Data Availability

Data are available upon request. However, there are legal restrictions and the EU General Data Protection Regulation (GDPR), the German Data Protection Laws, and the Bavarian Hospital law apply, so some requests may have to be declined partially or completely.

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
