# Peer review of "Differences and Similarities in the Pattern of Early Metabolic and Morphologic Response after Induction Chemo-Immunotherapy versus Induction Chemotherapy Alone in Locally Advanced Squamous Cell Head and Neck Cancer"

_cancers, 2022, doi:10.3390/cancers14194811_

Round 1
Reviewer 1 Report
The paper is generally well written and the results will be of interest to clinicians treating head and neck cancer.
A main criticism is that the ICIT patients are being compared to a historical group of IC patients however the characteristics are quite different between the groups - the ICIT patients had oral cavity/oropharynx primaries in 62%, were higher T stage and almost 40% HPV+ whereas the IC patients had hypopharynx/larynx primaries in nearly 83% and only 5% HPV+. This could be addressed by writing the paper as the outcomes in 2 distinct cohorts of patients with comparisons being hypothesis generating.
The authors could clarify that pathological response was confined to the primary site. Around 70% of patients were node positive so some description of the PET or CT/MRI response in nodes compared to primary site would be helpful.
A consort diagram of the flow of patients and timing of interventions may be helpful to make the methods clearer.
The discussion could be expanded if space permits and include comment on possible biological differences between HPV+ oropharyngeal tumours in the ICIT group compared to the IC patients, e.g. how many of the false positive ICIT patients (high residual SUVmax with CR on endoscopy) were HPV+?
p values are shown as 0.000 starting in line 160 and also line 167, 168 and elsewhere in the text and tables - these need to be checked.
The text describes other thresholds for PET response being no better than that used in the paper however Table 3 shows that adding SUV max post <6 seems to give a numerical advantage for NPV and PPV in ICIT patients, could the authors confirm this is not a significant improvement? Also Table 3 states resSUVmax <=50% for Beck et al 2022 although the test line 230 suggests should be <=40% - authors should clarify.
Author Response
We thank the reviewer for the interest in our manuscript and the friendly reception of our results.
We share the opinion that the patient groups differ in several characteristics in terms of tumor size, UICC-stage and proportion of HPV-associated tumors and cannot rule out that this will influence the results of the two treatment groups. In this respect, it is absolutely correct to describe the results as hypotheses-generating. An additional evaluation was carried out with the question of whether the two essential parameters resSUVmax and resDmax are distributed unequally in the unbalanced groups in both treatment options. This could essentially be ruled out. This can be seen as an indication that the unequal distribution of, for example, the T-stage does not affect the main results.
- We have corrected Table 1.
- We have inserted the evaluation in lines 206-210 and 2017 to 222.
- We have expressed the limitations of the study in lines 339-344
In fact, the evaluation refers solely to the primary tumor. The inclusion of the remission of the lymph nodes would be too confused because of the amount of data for each individual lymph node, especially since the remission is too different. Essentially, the response of the lymph nodes did not influence the therapeutic decision regarding the treatment of the primary tumor. In the case of tumor persistence in the lymph node after definitive treatment, a neck dissection was provided.
An evaluation of the lymph nodes was also omitted because no pathological preparations were available as a reference for pathological remission. An assessment of the response of lymphonods could be made a later , if we also gain insight into the development of the nuclear medicine parameters of other lymphatic and hematopoietic organs. This evaluation is complex and will follow.
- We have made an addition in the part „Material and Methods“ in lines 125-127
An figure of the mode of treatment and evaluation has been added to the „Material and Methods“ in lines 146-186.
The evaluation of patients with complete remission in a representative sample from the former tumor region according to ICIT despite constant/increasing SUVmax (resSUVmax >/= 0.8 was in 1/20 patients with HPV-associated oropharyngeal cancer and 4/31 patients without HPV-associated oropharyngeal cancer. As a result, there was no difference. We did not discuss this because of the small number of cases.
- We have added the case numbers in lines 285-287 for orientation
The p-values have been checked.
- The checked values were displayed in blue font
We took another look at the paragraph on the comparability of the criteria and specified our statement to the effect that the EORTC criterion is worse and the Beck 2022 criterion is equivalent, but not clearly better.
- The statement can be found from line 275.
Reviewer 2 Report
The present manuscript aims to assess the role of PET vs. CT/MRI for the evaluation of response to immunotherapy and chemotherapy in patients affected by head and neck cancer.
Interesting paper, but some major revisions are needed:
1) English should be revised
2) the criteria for the response to therapy with histopathology are incomplete. Please discuss the criteria used for the definition of complete or non compete response by biopsy
3) the population selected for the end-point are not well balanced, this would be a great limitation...please explain your choose and add in th limitation section
4) the sub-paragraph in the results entitled: "metabolic response....as a predictors..." is not completely clear, mainly the references and also the last two sentences. The authors are invited to modify them.
5) data for the cut-offs should be added in the ROC curves
6) please add some references or explain in the materials and methods the resSUVmax> 125% for the definition of pseudo progression. In the current form it is not clear.
7) the discussion does not localise the attention on the results of the present manuscript and therefore it should be rewritten or amplified in the content.
Minor comments:
1) please be sure that the acronyms and the words are correct (MRT, others)
2) why did the authors use the change in diameter for MRI/CT studies rather than to use the RECIST criteria?
Author Response
Reviewer 2
We also thank the reviewer for the effort in the review, the content-dense work and the benevolent and critical assessment of the manuscript.
The manuscript has been revised with regard to typos.
- The changes can be found in blue font
The sampling biopsies after the induction phase, especially after ICIT, was carried out with great care. The further intensity of the definitive therapy depended on the result of the CD8 cells. Only an increase in CD8 cells led to a de-escalation of the definitive therapy, so that extensive tiussue samples were obtained. In addition, the samples were carried out after induction by the same team that performed the initial endoscopy. The area of the sampling was found in the tumor board after reviewing the old ones and the PET after induction coordinated between the ENT doctors and the nuclear medicine specialists and radiologists. In the first patients who no longer had tumor cells despite high residual SUVmax values, the biopsy was repeated, which always led to the same result. The high rate of CR despite high SUVmax was the starting point for the publication. The samples were evaluated by two experienced pathologists and the findings were discussed in the tumor board.
- We have added remarks to the topic from line 140
The populations correspond to different temporal cohorts. In the group of IC patients, induction was used to make decisions between surgery and conservative therapy, in the ICIT group for decision-making between conservative forms of therapy due to more advanced tumor stages. Smaller tumors were not allowed to be included in the CHeckRad-CD8 study. As a result, there are only the two populations, which in turn differ only in the addition of immunotherapy and not in PEt diagnostics. For the addressed question of early response in metabolic imaging, there are no other comparison groups worldwide with this high number of patients. We considered the choice of a sample with comparable tumor properties to be methodologically more critical.
- We have discussed this as a critical point of the study from line 340-344
We have modified the discussion part with regard to the core statements on the development of the image parameters during the short therapy and the interpretation that distinguishes between IC and ICIT.
- Please see line 331-351.
We have revised the chapter "metabolic response.... as a predictors revised to represent the reference to the previous publications and renamed.
- The accuracy of predicting CR to IC and ICIT using several established metabolic values based on resSUVmax
- Please see line 265-276
The points for the cut-offs for the most frequently used criterion were written in the legend the ROC curves.
- Please see line 263-264.
For the definition of progression/remission, we have adhered to the definition of EORTC. A corresponding reference #15 has been inserted.
- Please see line 461-463
- Young H, Baum R, Cremerius U, Herholz K, Hoekstra O, Lammertsma AA, Pruim J, Price P. Measurement of clinical and subclinical tumour response using [18F]-fluorodeoxyglucose and positron emission tomography: review and 1999 EORTC recommendations. European Organization for Research and Treatment of Cancer (EORTC) PET Study Group. Eur J Cancer. 1999 Dec;35(13):1773-82. doi: 10.1016/s0959-8049(99)00229-4. PMID: 10673991.
In the RECIST assessment, the largest diameter of the primary tumor is evaluated. That is what we have done. The assessment of the lymph nodes included in the RECISt evaluation was not the subject of the study. It was not decisive for finding therapy. Therefore, the measurement method could not be declared as carried out according to RECIST.
- Please see line 121.
Reviewer 3 Report
This is an interesting study about early metabolic and morphologic response after induction chemo-immunotherapy or induction chemotherapy alone in locally advanced squamous cell head and neck cancer.
The paper is well written. However, some issues remain.
The authors stated that the ICIT and IC cohorts were comparable in terms UICC stage and tumor grade distribution. However, this was not true (p<0.05 in table 1).
P values must be exactly reported also when p>0.05. Moreover, if p is very low, please write p<0.001.
Tumor site must be reported and included in the analyses. Negative pathological specimens after one cycle of IC or ICIT are not reliable as marker of response to treatment. Indeed, submucosal cancer may remain and it was detected by imaging.
Comparison between IC and ICIT must take into consideration pre-treatment parameters. Therefore, adequate statistical analyses must be used.
CPS score must be reported.
Some graphics may help the readers.
Author Response
We would like to thank the reviewer for the very constructive comments, which partly coincide with those of the first two reviewers.
We share the opinion that the patient groups differ in several characteristics in terms of tumor size, UICC-stage and proportion of HPV-associated tumors and cannot rule out that this will influence the results of the two treatment groups. In this respect, it is absolutely correct to describe the results as hypotheses-generating. An additional evaluation was carried out with the question of whether the two essential parameters resSUVmax and resDmax are distributed unequally in the unbalanced groups in both treatment options. This could essentially be ruled out. This can be seen as an indication that the unequal distribution of, for example, the T-stage does not affect the main results.
- We have corrected Table 1.
- We have inserted the evaluation in lines 206-210 and 2017 to 222.
- We have expressed the limitations of the study in lines 339-344
We have focused on complete remission as a parameter for comparing the significance of the image properties, because it is still the most reliable pathological endpoint to be determined in the evaluation of remission if attention is paid to the type and extent of sampling.
We can assure you that the sampling after induction was extensive and deep and did not extend to the superficial areas. The sample was intended to determine the density of CD8 cells as a basis for deciding on further treatment. The biopsy was also repeated in the first patients who had little or no tumor cells left in the preparation. Ultimately, the experience of the IC group, in which a high SUVmax was almost always accompanied by a residual tumor, led to the decision that this was ultimately laryngectomized in the submitted publication. The intention of our work is the extent of the misinterpretation.
You asked for the presentation of the CPS-Scores. In this article we would like do not yet address the understandable question of the relationship between immunological parameters and image parameters due to complexity. We would like to process the few remaining tissue samples oft he IC group for the processing of a variety of immunological parameters, such as CPS, TPS, CD8, CD3, CD68 to compare it with the ICIT group. But we strongly agree that this is an extremely interesting question.
Round 2
Reviewer 1 Report
The authors have addressed issues raised by the reviewers.
Can the authors double check Figure 1, the box on the far left of the last line, should this be "No tumor cells in the biopsy (CR) n=32"?
Author Response
Thank you for the time and effort you put into the review.
Reviewer 2 Report
The manuscript is significantly improved and therefore it can be published.
Author Response
Thank you for your consideration of our research findings.
Reviewer 3 Report
Thank you for improving the manuscript.
Author Response
Many thanks for your effort.